# Sensory Processing in Children and Adolescents with Neurofibromatosis Type 1

**DOI:** 10.3390/cancers15143612

**Published:** 2023-07-14

**Authors:** Natalie A. Pride, Kristina M. Haebich, Karin S. Walsh, Francesca Lami, Melissa Rouel, Alice Maier, Anita K. Chisholm, Jennifer Lorenzo, Stephen J. C. Hearps, Kathryn N. North, Jonathan M. Payne

**Affiliations:** 1Kids Neuroscience Centre, The Children’s Hospital at Westmead, Sydney, NSW 2145, Australia; melissa.rouel@uts.edu.au (M.R.);; 2Murdoch Children’s Research Institute, Parkville, VIC 3052, Australiafrancesca.lami@mcri.edu.au (F.L.);; 3Department of Paediatrics, Faculty of Medicine, Dentistry and Health Sciences, University of Melbourne, Parkville, VIC 3010, Australia; 4Center for Neuroscience and Behavioral Medicine, Children’s National Hospital, George Washington University School of Medicine, Washington, DC 20052, USA; kwalsh@childrensnational.org; 5Royal Children’s Hospital, Parkville, VIC 3052, Australia

**Keywords:** neurofibromatosis type 1, sensory processing, autistic, ADHD, anxiety, adaptive skills, sensory modulation

## Abstract

**Simple Summary:**

Difficulties in sensory processing are often found in neurodevelopmental disorders and can significantly impact how a child responds to and functions within their environment. Studies examining sensory processing in children with neurofibromatosis type 1 (NF1) are sparse. This cross-sectional study aims to address this gap by examining parent-reported sensory processing in a sample of 152 children with NF1. Approximately 61% of children with NF1 displayed differences in how they respond to sensory stimuli when compared to a typically developing control group. These difficulties were seen equally across ages and sex and were found to be associated with a higher degree of autistic behaviors, ADHD symptoms, lower adaptive skills, poorer social skills, and increased anxiety and affective symptoms. The results highlight the importance of accommodating multisensory processing difficulties at home and school when deciding how to support a child with NF1 across environments.

**Abstract:**

Despite the evidence of elevated autistic behaviors and co-occurring neurodevelopmental difficulties in many children with neurofibromatosis type 1 (NF1), we have a limited understanding of the sensory processing challenges that may occur with the condition. This study examined the sensory profile of children and adolescents with NF1 and investigated the relationships between the sensory profiles and patient characteristics and neuropsychological functioning. The parent/caregivers of 152 children with NF1 and 96 typically developing children completed the Sensory Profile 2 (SP2), along with standardized questionnaires assessing autistic behaviors, ADHD symptoms, internalizing symptoms, adaptive functioning, and social skills. Intellectual functioning was also assessed. The SP2 data indicated elevated sensory processing problems in children with NF1 compared to typically developing children. Over 40% of children with NF1 displayed differences in sensory registration (missing sensory input) and were unusually sensitive to and unusually avoidant of sensory stimuli. Sixty percent of children with NF1 displayed difficulties in one or more sensory modalities. Elevated autistic behaviors and ADHD symptoms were associated with more severe sensory processing difficulties. This first detailed assessment of sensory processing, alongside other clinical features, in a relatively large cohort of children and adolescents with NF1 demonstrates the relationships between sensory processing differences and adaptive skills and behavior, as well as psychological well-being. Our characterization of the sensory profile within a genetic syndrome may help facilitate more targeted interventions to support overall functioning.

## 1. Introduction

Sensory processing refers to the ability of the nervous system to detect, modulate, and interpret sensory input (e.g., auditory, vestibular, visual, and tactile) and use this information to understand experiences and organize appropriate responses. One of most recognized models of sensory processing, Dunn’s Four-Quadrant Model of Sensory Processing [1,2], is often used to describe interindividual differences in sensory processing. The model, which is based on behavioral and neuroscientific data, proposes that individuals have unique neural thresholds for responding to sensory information, which, in turn, impact how one responds to their everyday environment. The sensory threshold is established as the pattern of interchange between habituation and sensitization to sensory stimuli. Individuals with low thresholds (hyper-responsive) have systems that are easily activated by sensory stimuli, thus they may be quick to respond. For example, individuals with tactile hyper-responsivity may avoid going barefoot or express distress when having their hair brushed. Those with high thresholds (hyporesponsive) may not be as responsive, demonstrating low stimuli awareness. In this instance, individuals with tactile hyporesponsiveness may have a reduced response to painful tactile stimuli or be unaware of another person’s touch, unless it is intense. Hyper- and hyporesponsiveness are not always mutually exclusive; a child can be hyporesponsive in one modality (e.g., auditory) and hyper-responsive in a different modality (e.g., visual) [3]. The second construct of Dunn’s model, behavioral response, exists on a continuum based on whether an individual has a passive or active strategy in responding to their environment. Individuals favoring passive strategies may internally respond to stimuli or ignore stimuli. Individuals favoring active strategies directly control the type and amount of sensory input in their environment. An individual’s sensory threshold and behavioral response can thus be categorized into four patterns of sensory processing: low registration (high threshold/passive response), hypersensitivity (low threshold/passive response), sensory seeking (high threshold/active response), and sensation avoiding (low threshold/active response).

Difficulties in sensory processing can significantly impact how a child responds to and functions within their environment. Sensory difficulties have been associated with challenging behaviors, anxiety, adaptive skills, social interactions, language development, motor performance, and academic achievement [4,5,6,7,8,9,10,11]. This impact is exemplified in individuals with autism spectrum disorder (hereon referred to as autism), where atypical responses to sensory stimuli are reported in over 70% of children [3,12]. Processing differences in children with autism are reported across sensory modalities, including tactile, visual and auditory domains [13], with both hypo- and hyper-responsiveness reported. These differences can have a considerable impact on social interactions and day-to-day functioning [8,14]. Hypersensitivity or hyposensitivity to sensory stimuli is considered a core symptom of autism as conceptualized by the Diagnostic and Statistical Manual of Mental Disorders 5th Edition—Text Revision (DSM-5-TR) [15], with some studies suggesting sensory processing difficulties, such as hyper-responsiveness to sound or touch, may be among the first indicators of autism [16,17].

There is increasing interest in better understanding sensory profiles in children with genetic syndromes associated with autism and other co-occurring conditions and to examine relationships between sensory processing and neurodevelopmental outcomes. Neurofibromatosis type 1 (NF1) is a common genetic condition arising from pathogenic variants in the *NF1* gene on chromosome 17 [18]. While typically classified as a tumor predisposition syndrome, the most common complications in children with NF1 are difficulties with social, behavioral, academic, and cognitive functioning [19,20,21,22,23]. Approximately 30–50% of children with NF1 meet criteria for attention-deficit hyperactivity disorder (ADHD) [24,25], up to half display autistic behaviors [26,27,28], and approximately 25% meet the diagnostic criteria for autism [29,30], all of which are associated with sensory processing difficulties [31,32,33]. Despite the high prevalence of these neurodevelopmental difficulties, studies examining sensory processing in children with NF1 are sparse. The recent evidence suggests differences in how infants with NF1 process auditory information, which has been associated with the later emergence of autistic traits [34]. Alterations in auditory processing [35], including the perception of the temporal characteristics of a sound [36], have also been detected in children with NF1, with some finding a relationship between these differences and functions, including the degree of language impairment and communication disorder [36] and phonological processing [35]. While the study of isolated sensory systems within a genetic condition provides valuable knowledge, multisensory integration is needed for many functions, including communication and language [37,38]. Increasing awareness of not only the range of modalities affected in NF1 but also the sensory profile and sensory-related behavioral responses will be important in informing and identifying effective sensory-based interventions for children with NF1.

The current study was designed to characterize the sensory processing profile of children with NF1. We addressed three main aims in this study: (i) to examine the proportion of individuals with NF1 exhibiting unusual responses to sensory stimuli compared to typically developing (TD) children; (ii) to examine sensory processing, including response styles and modality, of children with NF1 compared to a TD control group; and (iii) to examine the association between sensory processing characteristics and age, sex, and quantitative dimensional measures of intellectual functioning, autistic behaviors, ADHD symptoms, internalizing symptoms, social skills, and adaptive functioning.

## 2. Methods

### 2.1. Participants

Participants were drawn from an ongoing international prospective cross-sectional study examining the social functioning and autistic behaviors in children with NF1 [39]. Children and adolescents with NF1 were recruited from three neurogenetic centers: (1) The Neurofibromatosis Clinic at The Royal Children’s Hospital, Melbourne, Australia; (2) The Neurogenetics Clinic from The Children’s Hospital at Westmead, Sydney, Australia; and (3) the Gilbert Neurofibromatosis Institute at Children’s National Hospital, Washington, DC, USA. NF1 participants were diagnosed by a neurologist or clinical geneticist using the current NF1 diagnostic criteria [40,41]. Selection criteria for the larger cross-sectional study were (a) participants aged between 3–15 years; (b) participants living with at least one parent/caregiver who is fluent in English; (c) no evidence of symptomatic intracranial pathology that may impact on cognitive or behavioral functioning, such as an acquired brain injury or hydrocephalus (asymptomatic lesions, such as optic gliomas, were allowed); and (d) no visual or auditory impairments that would compromise the validity of psychometric testing. TD controls additionally had no history of neurological, genetic, or psychological problems, including developmental delay. A full description of the recruitment procedure and selection criteria are described in the study protocol [39]. The final sample for the current study consisted of 152 children with NF1 and 96 controls. This research was approved by the respective Human Research Ethics Committees of the Royal Children’s Hospital (HREC/16/RCHM/137), the Sydney Children’s Hospitals Network (HREC/16/SCHN/42), and the Institutional Review Board of the Children’s National Hospital (Pro00007045).

### 2.2. Procedure

All participants underwent face-to-face assessment at their respective hospitals. Intellectual functioning was estimated using the FSIQ composite from the Wechsler Intelligence Scale for Children Version 5 (WISC-V) [42] for 6–15 year-old children or the Wechsler Preschool and Primary Scale of Intelligence, Fourth Edition (WPPSI-IV), for 3–5 year-old children [43]. Parents/caregivers were also asked to complete several questionnaires about everyday cognitive, social, and functional skills.

### 2.3. Parent/Caregiver Questionnaire Measures

#### 2.3.1. Sensory Profile 2

Sensory Profile 2 (SP2) [2] is an 86-item parent/caregiver questionnaire designed to assess sensory function. Items are measured on a 5-point scale ranging from “almost always” to “almost never.” SP2 provides a four-quadrant sensory profile: sensory seeking, hypersensitivity, sensory avoiding, and low registration. It also provides six sensory domain scores, including auditory, visual, touch, oral, movement, and body position. Higher scores on sensory quadrant and sensory modality scores are associated with higher levels of difficulty processing sensory information. The test manual [2] provides a Normal Curve and Sensory Profile 2 Classification System, based on responses from a normative sample of children from the general population (*n* = 697). Raw scores were converted to Z scores based on normative data, and scores were classified based on Dunn’s classification system as “much less than others” (lower 2%), “less than others” (between 1 SD and 2 SD below the normative mean), “just like the majority of others” (±1 SD from the mean and accounting for 68% of the normative sample), “more than others” (between 1 SD and 2 SD above the mean), and “much more than others” (upper 2%).

#### 2.3.2. Autistic Behaviors and ADHD Symptoms

Autistic behaviors and ADHD symptoms were evaluated using well-validated, standardized parental questionnaires. The presence of autistic behaviors was measured using the parent version of the Social Responsiveness Scale, 2nd Edition (SRS-2) [44], a 65-item questionnaire assessing the social communication difficulties, restricted interests, and repetitive behaviors that are often present in individuals with autism. Raw scores were converted into an age- and sex-specific SRS-2 total T score using norms derived from the manual. ADHD symptoms were measured using the Conners ADHD Rating Scale [45] for children 3–5 years of age and the Conners 3 [46] for children aged 6–15 years. Inattentive scales from the CADS and Conners 3 were merged into a combined ADHD inattentive T score. Likewise, the Hyperactive/Impulsive Scales from the CADS and Conners 3 were merged into a combined ADHD hyperactive/impulsive T score. Higher T scores indicate more severe difficulties for both questionnaires.

#### 2.3.3. Functional and Mental Health Measures

Adaptive skills were assessed using the parent/caregiver version of the Adaptive Behavior Assessment System, 3rd Edition (ABAS-3) [47]. This measure assesses 11 essential everyday living skills (communication, community use, functional academics, health and safety, home living, leisure, motor, self-care, self-direction, and social) and provides an overall General Adaptive Composite (GAC). The GAC is an age- and sex-normed standard score, with lower scores indicating lower adaptive functioning. Social skills were assessed using the Social Skills Improvement System Rating Scales (SSIS-RSs) [48]. This parent/caregiver questionnaire provides an overall standard score composite of social skills. Lower scores on this composite indicate poorer social skills. Emotional and internalizing symptoms were assessed using the parent/caregiver version of the Child Behavior Checklist (CBCL) [49]. The DSM-oriented affective and anxiety subscales were used, which are considered “purer” measures of depression and anxiety, as the internalizing composite score includes somatic manifestations, which might be affected by NF1-related physical symptoms [30]. Age- and sex-normed T scores are provided, with higher T scores indicating more severe symptoms.

### 2.4. Data Analysis

Between-group comparisons were performed using independent samples *t*-tests for continuous data, presented with Cohen’s *d* effect sizes. Chi-square tests and relative risk statistics were performed to compare groups on categorical variables. Correlations between variables were tested using Spearman correlation coefficients. Between-group *d* were also compared to effect sizes based on published data of SP2 in children with ASD (*n* = 77) and ADHD (*n* = 87) aged 3–14 years of age [50]. Data were analyzed using Stata 17.0. All tests were controlled for type 1 errors using Bonferroni correction.

## 3. Results

The group characteristics and summary scores of parent-reported clinical and functional questionnaire measures are shown in Table 1.

### 3.1. Sensory Processing Clinical Cut-Offs

We first considered sensory processing from a categorical perspective, classifying children that demonstrated sensory problems exceeding the normal range (“more than others + 1SD to + 2SD” and “much more than others > 2SD”) to those that did not (Figure 1). Compared to the TD control group, a higher proportion of children with NF1 experienced sensory seeking (*p* = 0.001, 26.2% versus 8.7%, RR = 3.01, 95% CI 1.47–6.16), sensory avoidance (*p* < 0.001, 41.0% versus 7.4%, RR = 4.80, 95% CI 2.41–9.62), hypersensitivity (*p* < 0.001, 42% versus 4.3%, RR = 9.66, 95% CI 3.63–25.69), and low registration (*p* < 0.001, 46.0% versus 9.9%, RR = 4.65, 95% CI 2.43–8.87).

### 3.2. Sensory Processing in NF1 and TD Controls

Figure 2 shows NF1 and control group mean estimates and the between-group effect sizes for each sensory quadrant and modality. All sensory quadrants were significantly higher in children with NF1, with medium-to-large effect sizes. Significant group differences were also present for all sensory modalities, including auditory, touch, movement, body position, and oral senses, with the exception of the visual modality (Figure 2).

We next examined the number of sensory responses (quadrant scores) and modalities impacted for each child with NF1 (Figure 3a,b). Children with NF1 affected by unusual sensory processing were most likely to be affected in all four quadrants. Most children with NF1 and sensory processing difficulties were affected in either one or two modalities. However, some children with NF1 were affected by more, with 9.5% affected in five modalities and 5.4% in all six.

### 3.3. Comparison of SP2 in NF1 to Published ADHD and Autism Data

To compare the magnitude of sensory processing difficulties in children with NF1 to other neurodevelopmental conditions, the effect sizes of children with NF1 are displayed in Figure 4, with a comparison to the published data of children with autism and ADHD [50]. The data suggest similarities between all three groups in registration and between the NF1 and autistic cohorts for sensory seeking, but the autistic cohort demonstrates larger effects sizes than the NF1 group for sensitivity and avoidance.

### 3.4. Associations between Sensory Processing and Other Patient Characteristics and Functioning in Children with NF1

Next, we examined clinical and patient characteristics associated with sensory processing in children with NF1. To investigate whether variability in sensory differences was associated with other clinical features of NF1, effect sizes (ESs) were calculated between the main SP2 quadrant scores and the patient characteristics (sex ES = Cohen’s *d*; age, sex, FSIQ, SRS-2 total T score, ADHD symptom T score ES = Spearman’s rho). The ES *p*-value was adjusted for multiple correlations using Bonferroni correction. Neither sex nor age were significantly associated with the SP2 quadrant scores, indicating that sensory difficulties did not vary with age or differ based on sex (Table 2). Increased severity of sensory processing behaviors in all four quadrants (sensory seeking, low registration, hypersensitivity, and sensory avoidance) was moderately to strongly associated with a higher degree of autistic behaviors (SRS-2 total T scores), increased hyperactivity/impulsivity (ADHD hyperactive/impulsive T score), and increased inattention (ADHD inattentive T score). A significant but weak negative relationship was also found between FSIQ and sensory registration and sensory seeking (Table 2).

To assess the functional impact of sensory processing difficulties in children with NF1, Spearman rho correlations were conducted between the sensory quadrant scores and clinical outcomes. More severe sensory difficulties in all four quadrants were associated with lower adaptive functioning (ABAS-3 GAC), poorer social skills (SSIS total score), elevated anxiety (CBCL anxiety DSM subscale), and affective (CBCL affective DSM subscale) symptoms (all, *p* < 0.001, Table 2).

## 4. Discussion

This is the first study to delineate the profile of behaviors toward sensory stimuli in children with NF1 and their association to patient characteristics and functioning. Except for the visual modality, children with NF1 displayed higher levels of unusual responses to sensory stimuli across all dimensions of responsiveness and sensory modalities than the TD controls. The effect sizes were moderate to large. These difficulties were observed equally across the age range in the study (3–15 years) and across both sexes. The relative risk of developing sensory processing problems was 3 to 9.6 times greater for children with NF1 compared to the TD controls, with hypersensitivity found to be the greatest risk, followed by low registration, sensory avoidance, and sensory seeking. Approximately 61% of children with NF1 had one or more modalities impacted, with difficulties commonly occurring across several modalities and in several areas of responsiveness. These results suggest that if a child with NF1 experiences sensory processing challenges, it is likely these will occur across multiple areas of sensory processing. They may be sensory seeking, have low registration, avoid sensory stimuli, and, at the same time, be hypersensitive to sensory stimuli, as these patterns are not mutually exclusive [1]. For example, a child with NF1 may seek out tactile input by playing with textured materials, but, at the same time, have difficulty registering certain sounds. The same child may also be avoidant of particular textures, such as certain clothing, and sensitive to loud noises. When considering how to effectively support children with NF1, clinicians, parent/caregivers, and educators should take multisensory processing into account and note that each child’s sensory processing pattern is unique and may vary in different situations.

We observed a moderate-to-strong relationship between the SP2 scores and autistic behaviors and ADHD symptoms in the current study. This demonstrates that children with NF1 who experience sensory processing difficulties are also more likely to display autistic behaviors and ADHD symptoms and suggests that sensory processing difficulties are an important part of the NF1 neurodevelopmental phenotype. Our results also demonstrate a similar sensory processing profile in NF1 to those reported in children with ADHD and autism [8,32,50,51,52]. That is, all three groups display difficulties with sensory avoidance, sensitivity, registration, and seeking. While the magnitude of registration difficulties was similar between the three groups, there were slightly different degrees of severity between groups for sensitivity and avoidance. These similarities in sensory profiles may provide insight into shared underlying mechanisms. The sensory style of registration, for which a similar magnitude of registration difficulties is experienced in children with NF1, ASD, and ADHD [50,51,53], reflects the degree to which children orient or “tune in” to environmental stimuli [2]. A child with low registration may exhibit a lack of response to their name being called or may not respond or orient to social stimuli (faces) appropriately. In theory, a young child who does not overtly respond to novel sensory and social stimuli misses learning opportunities that are foundational to the development of social communication and adaptive skills. One potential explanation of registration difficulties (i.e., hyporesponsiveness) in children with NF1, which has also been suggested for children with ADHD and ASD [50,54,55,56], may be related to altered attentional mechanisms. Behavioral orienting is one measure of a child’s responsiveness to novel sensory information and is theorized to be driven by interacting “top down” dorsal frontoparietal and “bottom up” ventral frontoparietal attention networks [57]. These attention networks are hypothesized to be different in individuals with autism [58,59], potentially driving a reduced tendency to orient and attend to social stimuli. The evidence from a task-based functional magnetic resonance imaging (fMRI) study in children with NF1 indicates similar differences in the neural networks associated with orienting to sensory stimuli [60], suggesting abnormalities in these neural networks may lead to inefficient or faulty attentional orienting of sensory information. Resting-state fMRI may additionally prove a valuable tool in examining these neural networks and how they relate to shared behavior across diagnostic groups. The evidence indicates that each sensory domain (registration, seeking, sensitivity, and avoiding) is associated with a distinct intrinsic brain functional connectivity pattern [61]. Low registration, for example, has been associated with differences in connectivity in the frontoparietal and visual networks in children with ASD and ADHD [61]. While differences in functional connectivity have been observed in children with NF1 [62], future studies that examine the relationships between sensory responses and connectivity patterns between groups (NF1, ASD, and ADHD) may help us understand the overlap in sensory symptoms–neural circuits and ASD and ADHD symptom relationships across diagnoses.

Aligned with the current study, the electrophysiological studies of human brain activity have indicated sensory differences in NF1 are likely to occur across multiple sensory systems. Within vision, abnormal visual-evoked potentials and electroencephalography (EEG) responses have been reported, with findings suggestive of NF1-related differences in the later stages of visual processing and enhanced amplitude of alpha oscillations supporting deficits in basic sensory processing in NF1 [63]. More recently, Begum-Ali et al. (2021) [34] used EEG to measure auditory-evoked responses in infants with NF1. Relative to the controls, infants with NF1 demonstrated a prolonged latency in showing a differentiated neural response when detecting changes in auditory stimuli. This suggests an atypical response to auditory stimuli very early on in the development of children with NF1. Animal models of NF1 also provide a unique opportunity to study the cellular and molecular mechanisms underlying sensory difficulties in NF1. The recent behavioral and physiological data in drosophila have revealed that the loss of *Nf1* in peripheral sensory neurons leads to sensory processing errors. Importantly, these errors contribute to the impaired detection and/or processing of social cues, leading to social deficits in the mutant fly [64]. This suggests a disrupted flow of sensory information may contribute to downstream effects on behavior in NF1. Supporting this link between the *NF1* gene and sensory processing, Dyson et al. (2022) showed that the loss of *Nf1* in drosophila resulted in tactile hypersensitivity, which was associated with Ras-dependent synaptic transmission deficits indicative of neuronal hyperexcitability [65]. It will be important for future research to develop translation-relevant sensory processing outcomes, which bridge human and animal studies, to help better understand the causal pathways of these clinical manifestations.

This study’s findings have important implications for the management of children with NF1 and the development of interventions that aim to support their overall functioning. While the examination of anxiety and mood disorders in children with NF1 has been relatively neglected, our study and others [66,67] report increased anxiety and depressive symptoms compared to TD controls. While these symptoms are thought to be multifactorial in etiology, our results suggest a relationship between sensory processing difficulties and anxiety and affective symptoms, indicating they may co-occur in children with NF1. Sensory processing difficulties may have a role in understanding the heightened prevalence of anxiety and affective symptoms reported in NF1 and highlight the importance of supporting a child experiencing sensory processing differences. There were also several moderate-to-strong relationships observed between the sensory domains and functioning, including adaptive skills and social skills in the current study. These findings support the literature that reports a close association of sensory processing difficulties with functional limitations in daily life [1,9] and further highlight the importance in accommodating sensory processing difficulties at school and at home. Educators and clinicians should take multisensory processing into account when deciding how to support a child with NF1 across environments. Parents of children with NF1 who show sensory processing difficulties should be provided with information and resources about sensory processing behaviors to make sure that supportive strategies are implemented across contexts. Sensory processing interventions typically involve the child working with a clinician on strategies designed to retrain the senses, including auditory, visual, tactile, proprioceptive, oral, olfactory, vestibular, and interoceptive (the sense involved in the detection of internal regulation, such as heart rate and respiration). There continues to be some variability in the field regarding treatment and the theories underlying sensory interventions [68,69]. The evidence of the efficacy of these interventions in other developmental conditions, such as autism, is still a work-in-progress [68,69], with the overall evidence limited due to the lack of large-scale interventions studies (see [68] for review). Some positive effects, though, have been published supporting the Ayres Sensory Integration Intervention as an evidence-based intervention for sensory problems in autism [70,71]. This play-based individualized intervention uses sensory–motor activities that address specific difficulties identified in an assessment, which are linked to life functioning. Future research involving large-scale clinical trials within an evidence-based framework is warranted to determine the efficacy of this approach in general, as well as in children with NF1.

This study is not without limitations. While parent report offers an ecologically valid assessment of sensory processing in daily life, it is solely based on parental observations rather than the subjective experiences of the child. Collecting data from multiple informants, including the child, may provide a better understanding of sensory processing differences in children with NF1. Future studies that combine child-directed paradigms of sensory processing (e.g., low-level auditory processing or EEG) and behavioral assessment (SP2) may provide insight into how abnormalities in sensory systems may be related to sensory processing behavior profiles.

## 5. Conclusions

In summary, this study’s results indicate that children with NF1 are up to nine times more likely to experience sensory processing differences than their unaffected peers. Apart from vision, all sensory modalities are affected, and the NF1 sensory processing profile is broadly similar to children with autism and ADHD, showing avoiding, sensitivity, registration, and seeking difficulties, although to different degrees of severity. Sensory processing difficulties in children with NF1 may lead to behavioral alterations, such as poor eye contact, avoidance of noisy places, anxiety, and impaired social reciprocity, which may impact on everyday functioning. NF1 animal models also display deficiencies in sensory processing that are linked to social deficits, which may help us to understand the mechanisms of sensory hyper/hyposensitivity. As sensory deficits are relatively more tractable from circuit mechanisms and appear early on in NF1 development, the sensory domain holds promise for revealing mechanisms that may contribute to the development of higher-level behavioral difficulties in NF1, such as social impairment, but may also provide a translational platform to not just develop biomarkers but facilitate the ongoing search for new therapeutic approaches in NF1.

## Figures and Tables

**Figure 1 cancers-15-03612-f001:**
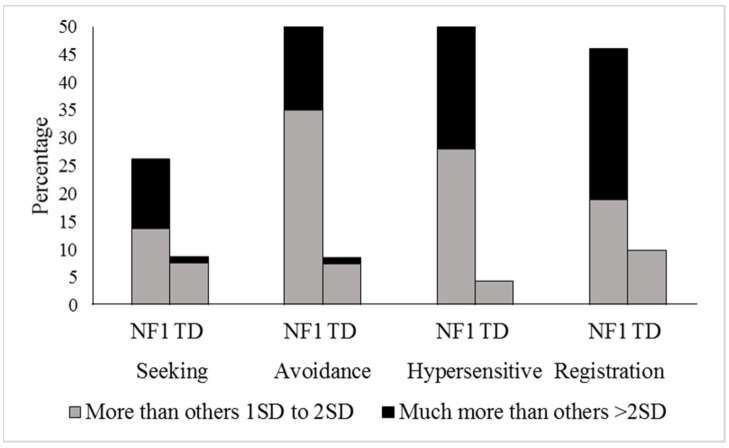
Percentage of NF1 and typically developing control participants who exceed the normal range on SP2.

**Figure 2 cancers-15-03612-f002:**
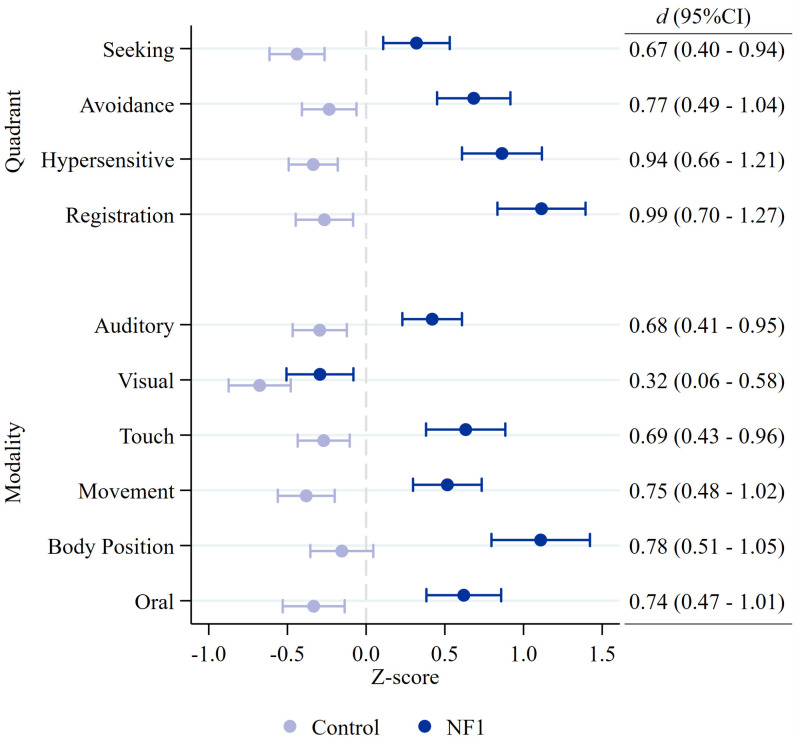
NF1 and control group mean estimates and between-group effect sizes (Cohen’s d) for each child Sensory Profile 2 quadrant and modality score. Circles indicate group mean estimate with 95% confidence intervals.

**Figure 3 cancers-15-03612-f003:**
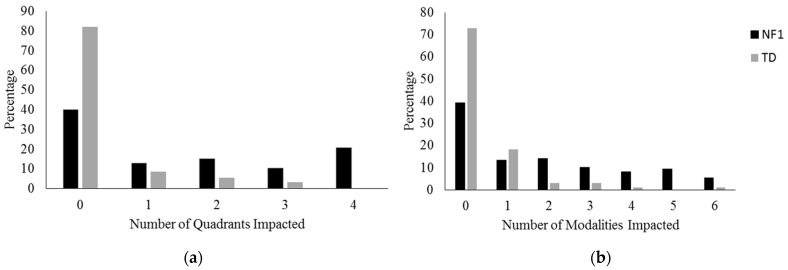
Number of atypical Sensory Profile 2 quadrants (**a**) and sensory modalities (**b**) for each child with and without NF1 (%).

**Figure 4 cancers-15-03612-f004:**
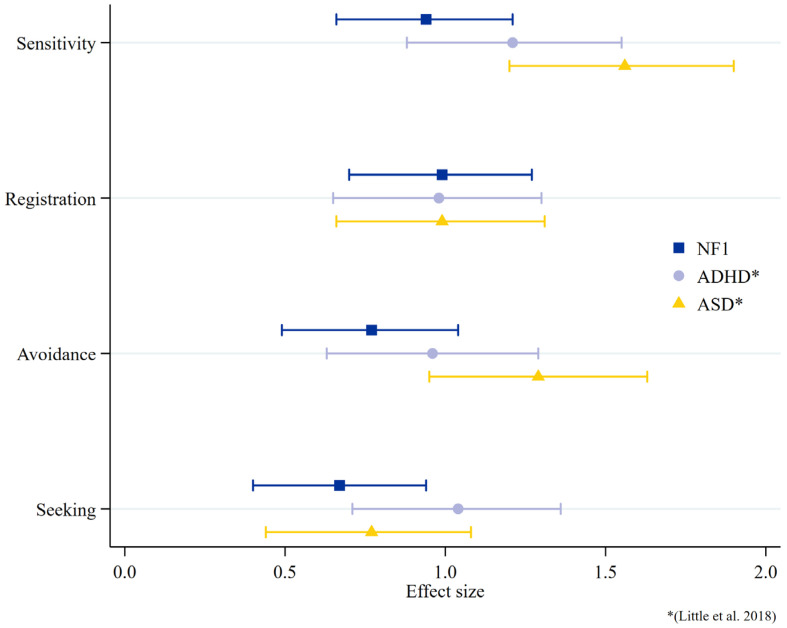
Group difference (NF1 > TD control) effect size on the Sensory Profile 2 (SP2) quadrant subscales. ASD and ADHD data are based on published SP2 data from Little et al., 2018 [50].

**Table 1 cancers-15-03612-t001:** Group characteristics and summary scores for clinical measures.

Variable	NF1 Mean (SD)	TD Controls Mean (SD)	F/χ^2^	*p*
Age (years)	8.1 (3.2)	8.1 (3.1)	0.001	0.982
Sex (% male)	54	49	0.59 ^a^	0.440
FSIQ ^b^	89.1 (12.7)	106.0 (17.8)	73.83	<0.001
SRS-2 Total ^c^	61.9 (14.1)	46.6 (5.8)	100.56	<0.001
ADHD Inattentive ^c^	64.9 (15.6)	52.3 (11.3)	46.74	<0.001
ADHD Hyperactive/Impulsive ^c^	64.0 (16.5)	51.4 (12.6)	40.91	<0.001
ABAS-3 GAC ^b^	86.3 (13.7)	101.7 (11.3)	79.92	<0.001
SSIS-RS Total ^b^	90.03 (17.5)	105.4 (13.7)	51.3	<0.001
CBCL Anxiety ^c^	57.7 (8.8)	53.2 (4.8)	52.85	<0.001
CBCL Affective ^c^	60.8 (9.0)	53.6 (5.2)	52.86	<0.001

^a^ Group differences analyzed using chi-square test. ^b^ Standard score (*M* = 100, *SD* = 15). ^c^ T score (*M* = 50, *SD* = 10). Note: ABAS-3 GAC = Adaptive Behavior Assessment System—3 General Adaptive Composite; ADHD = attention-deficit hyperactivity disorder; CBCL = Child Behavior Checklist; FSIQ = Full-Scale Intelligence Quotient; SRS-2 = Social Responsiveness Scale—2; SSIS-RS = Social Skills Improvement System Rating Scale.

**Table 2 cancers-15-03612-t002:** Spearman rho correlation coefficients between SP2 quadrant scores and clinical and functional outcomes in children with NF1.

	SP2 Quadrants			
	Sensory Avoiding	Hypersensitivity	Registration	Sensory Seeking
Age	0.07	0.01	−0.02	0.11
Sex	−0.19	−0.10	−0.10	−0.29
ADHD Inattentive	0.67 *	0.73 *	0.75 *	0.75 *
ADHD Hyperactive/Impulsive	0.62 *	0.69 *	0.68 *	0.75 *
FSIQ	−0.19	−0.22	−0.27 *	−0.31 *
ABAS-3 GAC	−0.51 *	−0.60 *	−0.50 *	−0.53 *
SSIS-RS Total	−0.57 *	−0.62 *	−0.55 *	−0.50 *
SRS-2 Total	0.70 *	0.79 *	0.72 *	0.72 *
CBCL Anxiety	0.61 *	0.53 *	0.42 *	0.48 *
CBCL Affective	0.67 *	0.66 *	0.56 *	0.56 *

* *p* < 0.001. Note: ABAS-3 GAC = Adaptive Behavior Assessment System—3 General Adaptive Composite; ADHD = attention-deficit hyperactivity disorder; CBCL = Child Behavior Checklist; ns = not significant; SP2 = Sensory Profile 2; SRS-2 = Social Responsiveness Scale—2; SSIS-RS = Social Skills Improvement System Rating Scale.

## Data Availability

The data set analyzed in the current study is available from the co-corresponding authors on reasonable request.

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
