# Peer review of "Sensory Processing in Children and Adolescents with Neurofibromatosis Type 1"

_cancers, 2023, doi:10.3390/cancers15143612_

Round 1

Reviewer 1 Report

A really outstanding manuscript addressing a significant issue in NF1 with objective data that is useful and well-displayed in the  figures and tables. 

The authors address the fact that they did not use parallel patient surveys - it is well documented in REINS literature that that parent and patient ratings frequently do not correlate - but this gives us future opportunities for study.

A very small point regarding the use of orient vs. orientate. It is a little thing that bugs me, but in the discussion text you used "orient" (line 317), but then followed it with "orientating" (line 325) rather than the more consistent "orienting". Please choose one.

Author Response

We thank Reviewer 1 for their comments. We have rectified the inconsistency in the manuscript. 

Reviewer 2 Report

Dear Authors, 

I congratulate you for this difficult topic and for this important study. NF1 is an incompletely deciphered genetic disease and unfortunately children with this condition are often misunderstood both by parents and unfortunately also by medical staff sometimes. Your study demonstrated that children with NF1 displayed higher levels of unusual responses to sensory stimuli across all dimensions of responsiveness and sensory modalities than TD controls. These results should be published and taken into account when a patient with NF1 is evaluated. The management of such a patient must be customized according to his profile of behaviors towards sensory stimuli.

Author Response

We thank Reviewer 2 for their comments.

Reviewer 3 Report

The authors present a study of sensory processing and deficits in NF1 children in correlation to helthy population. The results revealed that 40% of children with NF1 displayed differences in sensory registration (missing sensory input), were unusually sensitive, and unusually avoidant of sensory stimuli. 60% displayed difficulties in one or more sensory modalities. Elevated autistic behaviors and ADHD symptoms were associated with more severe sensory processing difficulties.

The authors concluded that their detailed assessment of sensory processing, alongside other clinical features demonstrates relationships between sensory processing differences and adaptive skills, behavior, as well as psychological well-being. They postulated that the characterization of the sensory profile within a genetic syndrome may help facilitate more targeted interventions to support overall functioning. 

the manuscript is well written and the methods and material is good. The statitics are adequate. I suggest to address to following comment to improve the manusript before it might be published:

1. please add some information about correlation to neuroelectrophysiological testing and the sensory defitis in these cohort. and discuss.

2. are there therapeutical consequences? please add information and discuss.

Author Response

1. Please add some information about correlation to neuroelectrophysiological testing and the sensory deficits in these cohort and discuss.

We thank Reviewer 3 for this valuable suggestion. We have now added text regarding electrophysiological studies and sensory processing in NF1 the Discussion.

2. Are there therapeutical consequences? Please add information and discuss.

We also thank Reviewer 3 for this comment. While we feel we have addressed the therapeutic consequences of the current findings within the Discussion, we have now reinforced this in the Conclusion.